# HAZARD CHALLENGE: EMBODIED DECISION MAKING IN DYNAMICALLY CHANGING ENVIRONMENTS

**Qinhong Zhou**[1]*,**Sunli Chen**[2]*, **Yisong Wang**[3], **Haozhe Xu**[3], **Weihua Du**[2],
**Hongxin Zhang**[1], **Yilun Du**[4], **Joshua B. Tenenbaum**[4], **Chuang Gan**[1,5]
[1]University of Massachusetts Amherst, [2] Institute for Interdisciplinary Information Sciences,
Tsinghua University, [3]Peking University, [4]MIT, [5]MIT-IBM Watson AI Lab

## ABSTRACT

Recent advances in high-fidelity virtual environments serve as one of the major driving forces for building intelligent embodied agents to perceive, reason and interact with the physical world. Typically, these environments remain unchanged unless agents interact with them. However, in real-world scenarios, agents might also face dynamically changing environments characterized by unexpected events and need to rapidly take action accordingly. To remedy this gap, we propose a new simulated embodied benchmark, called HAZARD, specifically designed to assess the decision-making abilities of embodied agents in dynamic situations. HAZARD consists of three unexpected disaster scenarios, including fire 🔥, flood 🌊, and wind 🍃, and specifically supports the utilization of large language models (LLMs) to assist common sense reasoning and decision-making. This benchmark enables us to evaluate autonomous agents' decision-making capabilities across various pipelines, including reinforcement learning (RL), rule-based, and search-based methods in dynamically changing environments. As a first step toward addressing this challenge using large language models, we further develop an LLM-based agent and perform an in-depth analysis of its promise and challenge of solving these challenging tasks. HAZARD is available at https://vis-www.cs.umass.edu/hazard/.

## 1 INTRODUCTION

Embodied agents operate in a dynamic world that exhibits constant changes. This world experiences various changes at every moment, including the rising and setting of the sun, the flow of rivers, weather variations, and human activities. To successfully navigate and function in such an ever-changing environment, robots are required to *perceive* changes in their surroundings, *reason* the underlying mechanisms of these changes, and subsequently *make decisions* in response to them.

To simulate a dynamic world, it is necessary to create environments that can spontaneously undergo changes. Currently, various simulation platforms have emerged in the field of embodied AI, including iGibson (Shen et al., 2021), Habitat (Savva et al., 2019), SAPIEN (Xiang et al., 2020b), Virtual-Home (Puig et al., 2018), AI2THOR (Kolve et al., 2017), ThreeDWorld (TDW) (Gan et al., 2021), etc. Existing tasks on these simulation platforms involve agent exploration and agent-driven interactions, but they lack support for environment-driven changes, which are rather influential and unpredictable. The iGibson 2.0 (Li et al., 2021) platform partially supports spontaneous environmental changes to a limited extent, but these changes are limited to the propagation of a few variables between individual objects.

In this paper, we propose the HAZARD challenge, an innovative exploration of embodied decision-making in dynamic environments, by designing and implementing new capabilities for physical simulation and visual effects on top of the ThreeDWorld. HAZARD manifests itself in the form of unexpected disasters, such as fires, floods, and wild winds, and requires agents to rescue valuable items from these continuously evolving and perilous circumstances.

The HAZARD challenge places agents within indoor or outdoor environments, compelling them to decipher disaster dynamics and construct an optimal rescue strategy. As illustrated in Figure 1, the

---

*Qinhong Zhou and Sunli Chen contribute equally.

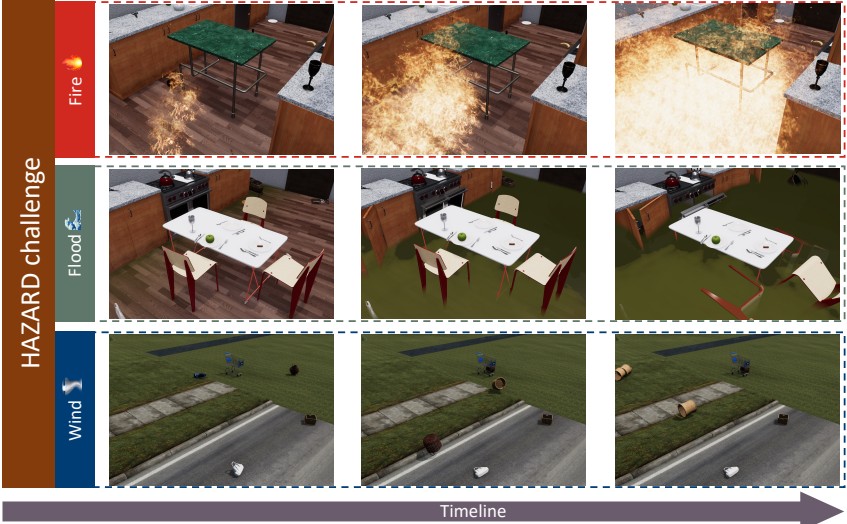

Figure 1: **Illustration of HAZARD Challenge.** The HAZARD challenge consists of three dynamically changing scenarios: fire 🔥, flood 🌊, and wind 🍃. In the fire scenario, flames continuously spread and burn objects. In the flood scenario, water spreads and rises, washing away objects and causing damage to non-waterproof objects. The wind scenario poses the challenge of objects being blown away, making them hard to reach. These scenarios present embodied agents with complex perception, reasoning, and planning challenges.

scenarios vary in severity and complexity. An indoor fire scenario might involve the rapid spread of flames, threatening flammable target objects. In an indoor flood scenario, an overwhelming volume of water inundates the house, jeopardizing non-waterproof targets. In an outdoor wind scenario, strong winds scatter lightweight objects across roads, making retrieval a challenging task for agents. To successfully rescue target objects from these disasters, agents must effectively transfer them to safe zones such as backpacks or shopping carts.

To facilitate this endeavor, we introduce a comprehensive benchmark comprising these disaster scenarios, complete with quantitative evaluation metrics. We also provide an API to employ large language models (LLMs) for action selection. This API integrates visual observations and historical memories into textual descriptions, thereby providing a semantic understanding of the dynamic environment. To optimize the use of LLMs, we compress a large volume of low-level actions by A* algorithm, significantly reducing the frequency of LLM queries.

We evaluate both LLM-based agents and several other decision-making pipelines on our benchmark, including a rule-based pipeline that operates based on a simple set of rules, a search-based pipeline that utilizes the Monte Carlo tree search (MCTS) algorithm for action selection, and a reinforcement learning-based pipeline. Through our experiments, we find while the LLM pipeline is capable of understanding and considering certain basic factors, such as object distance, it may encounter challenges in comprehending and effectively handling more complex factors, such as the dynamic nature of environmental changes.

The main contributions of our work are: 1) designing and implementing a new feature that enables the simulation of complex fire, flood, and wind effects for both indoor and outdoor virtual environments in TDW; 2) developing a comprehensive benchmark, HAZARD, for evaluating embodied decision-making in dynamically changing environments, as well as incorporating the LLM API into our benchmark; and 3) conducting an in-depth analysis of the challenges posed by perception and reasoning for existing methods, especially LLM-based agents in tackling the proposed benchmark.

## 2 RELATED WORK

**Simulators for Embodied AI** The recent advance of embodied AI has largely been driven by the development of simulation platforms. While earlier platforms primarily focused on supporting agent exploration (Savva et al., 2017; Beattie et al., 2016; Savva et al., 2019; Yi et al., 2018; Das et al.,

2018), recent platforms (Gan et al., 2021; Xiang et al., 2020a; Shen et al., 2021; Szot et al., 2021; Li et al., 2021; Puig et al., 2018; Kolve et al., 2017; Yan et al., 2018) have advanced by enabling physical agent-driven interactions. In this paper, we specifically focus on the impact of environment-driven changes on embodied agents, which remains a relatively unexplored area of research. Earlier works either supported spontaneous changes in the environment within limited ranges (Li et al., 2021), different environmental impacts on agent actions (Zeng et al., 2022), or just focused on identifying such changes, which occurred only during each reset (Landi et al., 2022).

**Embodied AI with Large Language Models** Recently, large language models (LLMs) (Brown et al., 2020; Chowdhery et al., 2022; Touvron et al., 2023a; Ouyang et al., 2022) have made remarkable strides in the field of AI, and their potential in embodied AI tasks has also been widely investigated and explored (Kant et al., 2022; Wake et al., 2023; Shah et al., 2023; Vemprala et al., 2023; Lin et al., 2023; Yang et al., 2023; Liu et al., 2023a). LLMs are capable of providing contextual information (Ahn et al., 2022), building inner monologues (Huang et al., 2022), providing model initializing weights (Li et al., 2022), and error correction (Wang et al., 2023) to enhance the planning of embodied agents. More directly, LLMs can generate policy code (Liang et al., 2022) or produce plans for embodied agents (Song et al., 2022; Driess et al., 2023). Different from previous works, we explore the planning and decision-making ability of LLMs in dynamically changing environments. We also implement APIs for LLM-based agents in HAZARD, providing support for future research in this area.

**Search and Rescue** Search and rescue (SAR) is a widely explored area for robotics. Most of the existing projects and programs in robot SAR focus on using unmanned aerial vehicles (Ollero et al., 2005; Mehmood et al., 2018; Merino et al., 2005; Hayat et al., 2016; Shakhatreh et al., 2019; Yeong et al., 2015), unmanned ground vehicles (Chikwanha et al., 2012; Santos et al., 2020; Cubber et al., 2017), and unmanned underwater vehicles (Binney et al., 2010; Zeng et al., 2015; Li et al., 2018) for searching one or more target in mostly static scenes. Some previous works also provide a dynamic simulation of disasters but are restricted to 2D abstract environments (rob, 2023) or limited scenes (Tang et al., 2018). Different from these works, we provide simulation for dynamically changing environments in embodied scenes and focus on decision-making in these environments.

## 3 THE HAZARD CHALLENGE

### 3.1 OVERVIEW

The HAZARD challenge sets itself apart through its distinctive dynamic scenarios, requiring high-quality physical and visual simulation of significant environmental changes such as fire, flood, and wind. To realize our objectives, we have developed an environment change simulation framework, premised on the robust foundation of the ThreeDWorld platform. This framework offers a reliable physical simulation system, with a versatile renderer capable of producing visual effects tailored to each type of environmental change. In this section, we delve into the implementation details of both how we simulate these dynamic environment changes and our new dataset.

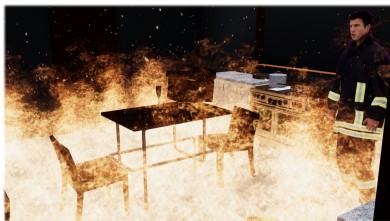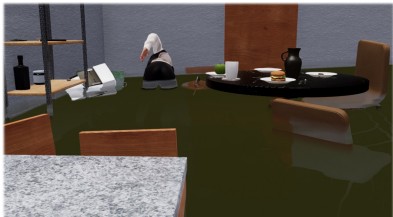

Figure 2: **Detailed visual effect of fire 🔥 and flood 🌊.** We have developed near-realistic visual effects for the fire and flood scenarios, which are controlled by our simulation system.

### 3.2 SCENES

#### 3.2.1 FIRE

In the fire scenario, we simulate an indoor scene of a room on fire. This dynamic environment presents various changes to the agent, including the spreading visual effects of the fire, changes in

temperature, and the transformation of objects as they burn or become burnt. To capture these effects accurately, we implement a temperature system and integrate it with the visual effects using the ThreeDWorld simulator.

**Temperature** The temperature system simulates the heat transfer process among all objects in the environment. According to the second law of thermodynamics, in this system, heat is transferred from objects with higher temperatures to those with lower temperatures. However, it is infeasible to simulate real heat transfer to a fine-grained level due to computational limits. Instead, we adopt an expedient approach described below.

For each object $o$ in each time frame, we update the temperature $T'(o) = T(o) \cdot (1 - d) + d \cdot T_{env}(o)$ where $d$ is the decay rate. $T_{env}(o)$ simulates the heat transfer from the environment and is defined as the weighted average of room temperature $T_{room}$ and surrounding objects $T(o')$. The weight for $T_{room}$ is a pre-defined constant $W_{room}$ and the weight for $T(o')$ is $W_{o'} = \min(D^{-2}, dist(o, o')^{-2})$ where $D$ is the distance threshold and $dist(o, o')$ is the distance between objects $o$ and $o'$. In summary, we have pre-defined constants $W_r, T_{room}$, decay rate $d$ and distance threshold $D$ to simulate real-life heat transfer between objects.

Not only the objects can catch on fire. In our setting, we divide the floor into 2-D grids where each grid can burn separately. Since the grid size is set small, it's infeasible to assign a temperature to each grid and update them every frame. Instead, we use a spreading model where a fire having burnt for time $t$ has probability $p(t)$ to spread to a nearby grid. $p(t)$ is a linear function so that after sufficient time the fire must spread.

**Object status and visual effect** To simulate the visual effects generated by a spreading fire, we define three distinct object statuses similar to (Li et al., 2021): normal, burning, and burnt. Objects in the normal status exhibit no additional visual effects. An object becomes burning once its temperature reaches the ignition point. As illustrated in Figure 2, burning objects are adorned with a fire visual effect on the top, with the scale of this visual effect gradually amplifying as the object combusts over subsequent frames. After a designated burning duration, an object becomes burnt and has a black hue to represent the burnt state.

### 3.2.2 FLOOD

The flood scenario is designed to simulate the spread and rise of water within an indoor room. This scene introduces the following changes to the agent: (a). The flood gradually submerges the objects within the scene, making them challenging to recognize and causing damage to non-waterproof objects, and (b). objects with low density have the tendency to float on the flood surface and may be swept away by the force of the flood.

**Visual effect** As Figure 2 shows, the flood surface is designed to be translucent, allowing the agent to perceive both the flood surface itself and the objects located beneath it. To simulate the spread of the flood, we rotate the surface into a sloped position and make it gradually rise and move forward. This combination of visual effects accurately depicts the dynamic nature of a spreading flood.

**Physical simulation** For each object in the flood, we incorporate two forces to simulate the physical effects of the flood: buoyancy and drag force. The buoyancy force, denoted as $F_B$, acts in the opposite direction to gravity and its magnitude is determined by $F_B = \rho_f V g$, where $V$ denotes the product of the volume of the submerged portion of the object, $g$ denotes the gravitational acceleration, and $\rho_f$ denotes the density of the flood. On the other hand, the drag force, denoted as $F_D$, is calculated using the drag equation $F_D = \frac{1}{2}\rho_f v^2 C_D A$, where $v$ represents the relative velocity of the object with respect to the fluid, $C_D$ denotes the drag coefficient, and $A$ represents the vertical area of the object. The direction of $F_D$ is opposite to the relative speed of the object in relation to the fluid.

### 3.2.3 WIND

In contrast to the fire and flood scenarios, the wind scenario simulates an outdoor scene where objects are affected by intense and turbulent winds. As a result, the primary dynamic feature of this scenario is the movement of objects induced by powerful wind forces. In real life, determining wind forces is a complex topic in aerodynamics, so we take an ideal model. Specifically, we assume the wind has a fixed velocity everywhere and objects have a face vertical to this velocity. With some physics

derivation, The force is $F = \rho_a v^2 A$ where $\rho_a$ is the air density, $v$ is the relative wind velocity and $A$ is the vertical area facing the wind. We modify it by setting $F = F_1 + F_2$ in which $F_1 = \rho_a v^2 A$ and $F_2 = \mathbf{r} \times F_1$. $\mathbf{r}$ is a random vector of fixed length so that $F_2$ simulates a random turbulence minor compared to $F_1$. We also apply a random torque to each object to account for forces applied not on the center of mass. The magnitude of forces and torques is hand-adjusted to create a dynamically changing scene.

To reduce the difficulty of the HAZARD challenge, environmental impacts like temperature, flood forces, and wind forces do not affect agents in the default setting. Additionally, we explore a more challenging scenario where agents are influenced by these environmental effects in Appendix B.

### 3.2.4 PROCEDURAL SCENE GENERATION

Based on the ProcGenKitchen pipeline from ThreeDWorld simulator, we develop a procedural generation pipeline tailored for the HAZARD challenge. This pipeline enables the generation of diverse scenes with dynamic changes. Initially, a basic scene is generated using ProcGenKitchen. Subsequently, we introduce additional elements to this basic scene in a random manner, including (a). target objects and additional objects on both floor and surfaces of the existing objects, such as tables and chairs, (b). agents' initial positions and directions, and (c). initial positions of the fire sources. This procedural generation pipeline ensures the creation of various dynamically changing scenes for the HAZARD challenge.

### 3.3 BENCHMARK DETAILS

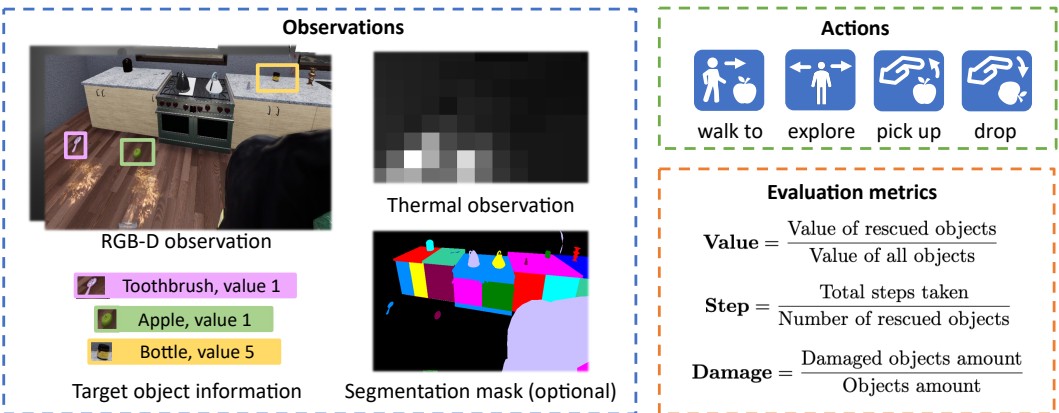

Figure 3: **Benchmark details**. In HAZARD challenge, an embodied agent needs to rescue a given set of objects from disasters. The agent observations include RGB-D signals, temperature or water level signals, target object information, and segmentation masks. To address the challenge in perception, we also provide a perceptional version of HAZARD which excludes segmentation mask from observations. The action space consists of four high-level actions: *Pick Up*, *Explore*, *Drop*, and *Walk To*, each representing a compression of multiple low-level actions. The final performance of agents is measured by 'value', 'step', and 'damage'.

**Problem definition** In the HAZARD challenge, the objective for an embodied agent is to rescue a predetermined set of objects, denoted as *target objects*, and bring them to a given safe location such as a bag held by the agent or a shopping cart. As Figure 3 shows, at each step, an agent receives an RGB-D observation, semantic segmentations of the observation, and a blurred environment-specific observation (temperature in the fire scenario and water level in the flood scenario). We also provide a perceptional version of HAZARD which excludes semantic segmentation from inputs. Given these observations, agents must make appropriate choices from the available *agent action space* at each step to accomplish the mission.

**Target objects** The objects that agents are required to rescue are denoted as "target objects." These objects are randomly placed on the floor or other surfaces within the environment. In each fire or flood scene, we randomly select 4 categories of objects from a universal target category pool, which consists of 22 object categories for fire or flood scenarios, and 11 object categories for wind scenarios. All objects falling within these selected categories are considered target objects. The

target category pool encompasses objects that exhibit diversity across four attributes: object value, waterproof capability, ignition point, and susceptibility to the wind.

**Agent action space** In our study, we utilize ThreeDWorld Replicant, a humanoid agent with basic actions such as *Move Forward By*, *Turn By*, *Reach For*, and *Reset Arm*. To accomplish the proposed task, we introduced compressed actions derived from these foundational actions. Specifically, the *Explore* action combines several *Turn By* actions, enabling agents to swiftly perceive their environment. A robust *Pick Up* action, formed by *Reach For* and *Reset Arm* actions, allows the agent to grasp target objects effectively. Additionally, a *Drop* action helps the agent put down held objects. Therefore, once the agent reaches the object, it can utilize the *Pick Up* action to hold the object and subsequently use the *Drop* action to position it in a safe location. HAZARD also supports direct utilization of low-level actions, including 'move_by', 'turn_by', and 'turn_to'.

**Support for LLM-based pipelines** The proposed benchmark also supports using LLMs as decision-makers. However, a significant challenge arises when querying the LLM for actions at each step, as it can lead to frequent queries and subsequently lower inference speed. To solve this problem, we implement agent navigation to objects using the A* algorithm. This navigation algorithm compresses a large number of *Move Forward By* and *Turn By* actions into a single *Walk To* action. As a result, apart from the *Pick Up* and *Drop* actions, the LLM is only queried to select which objects to *Walk To*. Based on this design, we implement efficient APIs for LLM-based pipelines in HAZARD.

**Evaluation metrics** For HAZARD challenge, we use *rescued value rate (Value)*, *averaged rescue step (Step)*, and *averaged damaged rate (Damage)* as evaluation metrics. The *rescued value rate* is calculated as the ratio of the total value of the rescued objects to the total initial value of all target objects. Objects that are damaged due to environmental changes will lose half of their value. Specifically, in the fire scenario, objects lose their value once they start burning. In the flood scenario, a non-waterproof object loses its value if it becomes submerged by the flood. The *averaged rescue step*, as a measurement of efficiency, is defined as the averaged step number to rescue an object for an agent. To improve the efficiency of the evaluation process, we set a time limit of 1,500 frames for tasks fire and flood while a longer 3,000 frames is set in the wind scenario. The *averaged damaged rate (Damage)* is calculated as the proportion of damaged objects among the objects that the agent rescued. Note that objects can only be damaged in a fire or flood scenario.

## 4 BUILDING LLM-BASED PIPELINE FOR EMBODIED AGENTS

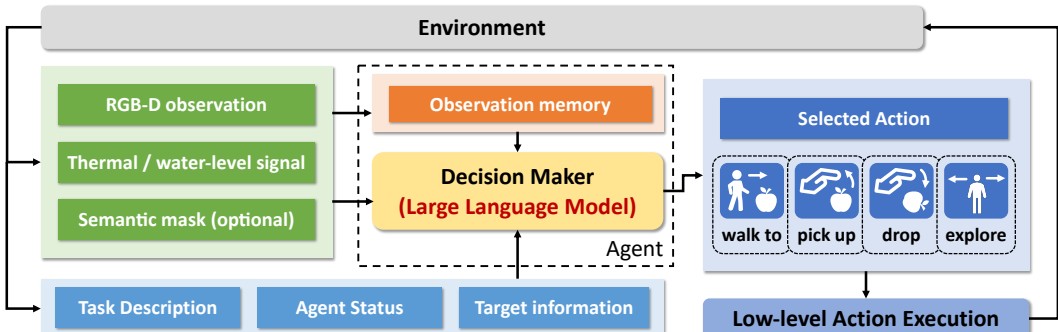

Figure 4: **Framework of the proposed LLM-based pipeline**. The LLM takes in diverse input information from the environment and engages in comprehensive decision-making. At the high level, the LLM selects actions such as "walk to", "pick up", "drop", or "explore". These chosen actions are then executed through a series of low-level actions. This hierarchical approach enables the LLM to effectively process and utilize the various types of input information available.

Taking inspiration from recent research (Song et al., 2022; Driess et al., 2023), we employ an LLM as the decision maker to determine the target destination for embodied agents. As illustrated in Figure 4, to enable the perception of LLM, we convert the information from the environment into text descriptions. Then LLM is required to select a proper action, which is subsequently converted into multiple low-level actions for execution. We provide the LLM decision maker in this step with the following information:

- **Task description**: The first part of the prompt consists of a manually designed description of the current task. The description briefly introduces the challenges the disaster agent needs to face, the overall goal of the agent, and the format of the following prompt parts.
- **Target information**: After the task description, we provide target information for LLMs, including the names, values, and properties of target objects.
- **Current state**: To provide LLM with comprehensive information about the current state, we convert visual observations, thermal or water-level signals, and semantic masks into a 2D semantic map. For the perceptional version of HAZARD, input semantic mask is replaced with a segmentation proposal provided by a perception model. Then the LLM is provided with a textual description of the semantic map, including object distance, temperature, water level, and value.
- **Observation memory**: To help LLMs infer the dynamics of the environment and predict future changes, we have designed an observation memory to store the historical states. During inference, LLMs utilize the observation memory by incorporating the description of each historical state into the prompt.
- **Available actions**: The prompt concludes with a list of the currently available actions. Therefore, LLMs make decisions by solving a multi-choice problem.
- **Other information**: The prompt also includes other necessary information, such as the agent's history of actions taken and its current status.

The detailed format of the prompt can be referred to as shown in Figure 7 in the Appendix.

## 5 EXPERIMENTS

### 5.1 EXPERIMENTAL SETUP

**Traning and testing setup** To create the dataset for HAZARD, we choose 4 distinct indoor rooms for the fire and flood tasks, and 4 outdoor regions for the wind task. Within each room or region, we generate 25 diverse scenes using our procedural generation pipeline (described in Section 3.2.4). One indoor room and one outdoor region are selected as the test set. As a result, we obtain a total of 100 unique scenes for each task, with a train-set split ratio of 3:1.

**Details of language model backbone** We evaluate the LLM-based agent in Section 4 with 3 different LLM backbones, including Llama-13b-chat model (Touvron et al., 2023b), OpenAI GPT-3.5-turbo (August 3 Version), and OpenAI GPT-4. We use max tokens of 512, temperature of 0.7, top p of 1.0 as hyper-parameters during inference.

**Perception Module** To get semantic segmentations in the perceptional version of HAZARD, we use OpenMMLab detection framework (Chen et al., 2019) to implement our perceptional model. We collect 200 images with ground truth segmentation in each training instance and use the collected data to fine-tune a Mask-RCNN (He et al., 2017) model provided by OpenMMLab.

### 5.2 BASELINES

We implement several baseline agents for evaluations as follows.

**Random agent** An agent randomly selects low-level actions to execute. To emphasize the hardness of our challenge, we provide more informative actions including walking to the nearest target object (container), picking up (dropping) the nearest object and exploring.

**RL model** We also trained reinforcement learning models using Proximal Policy Optimization (PPO) (Schulman et al., 2017). The actions are the same as described in the random agent. We design the function that rewards picking up and dropping correctly while penalizing actions that fail or have no effect. To make the reward function smoother, we also add a factor of the distance from the agent to the nearest object.

**Rule-based agent** An agent randomly chooses a target object to rescue. After selecting the target object, the agent automatically walks to the object, picks it up, and drops it into a safe place.

**MCTS agent** Monte Carlo Tree Search (MCTS) (Kocsis & Szepesvári, 2006) is a commonly used algorithm in decision-making problems, which has an effective balance of exploration and

exploitation. Since it is hard to get the ground truth frame costs of each action, we design several kinds of heuristic costs for MCTS, such as navigation heuristics, grasp heuristics, drop heuristics, and exploration heuristics. After that, we use MCTS to find an action plan with minimal total cost.

**Greedy agent** A simple greedy agent that persists in rescuing the nearest target object with the lowest cost. This agent chooses actions randomly when there are no target objects in observation or memory.

## 5.3 EXPERIMENTAL RESULTS

Table 1: The *rescued value rate (Value)*, *averaged rescue step (Step)*, and *averaged damaged rate (Damage)* of the proposed LLM pipeline (LLM) and all baseline methods. *Without perception* denotes the scenario that includes the semantic mask in the input, while the *with perception* scenario excludes the semantic input and requires the agents to perceive the environment with a perception model.

| Methods | Fire 🔥 | | | Flood 🌊 | | | Wind 🎐 | |
|---|---|---|---|---|---|---|---|---|
| | Value↑ | Step↓ | Damage↓ | Value↑ | Step↓ | Damage↓ | Value↑ | Step↓ |
| Without Perception | | | | | | | | |
| Greedy | 35.4 | 315.8 | 25.9 | 18.5 | 289.9 | 80.3 | 0.2 | **444.0** |
| Random | 43.8 | 279.1 | 37.3 | 28.1 | 286.6 | 80.0 | 7.1 | 1131.8 |
| Rule | 53.1 | 236.1 | 32.3 | 27.3 | 325.3 | 82.2 | 0.0 | - |
| RL | 46.1 | 277.3 | 33.6 | 35.0 | 252.5 | 71.7 | 12.4 | 889.5 |
| MCTS | 75.9 | **150.1** | 19.7 | 43.7 | 146.6 | 69.9 | 18.0 | 898.0 |
| LLM (Llama-13b) | 70.2 | 173.8 | 24.0 | 42.6 | 179.6 | 71.2 | 9.6 | 1255.6 |
| LLM (GPT-3.5) | 70.9 | 170.4 | 20.3 | 44.3 | 156.6 | **63.7** | 23.5 | 735.0 |
| LLM (GPT-4) | **77.8** | 159.9 | **15.9** | **45.7** | **142.9** | 64.9 | **31.1** | 590.1 |
| With Perception | | | | | | | | |
| Greedy | 35.5 | 257.8 | 25.3 | 21.5 | 250.7 | 68.8 | 0.2 | **442.0** |
| Random | 41.3 | 314.6 | 31.6 | 26.7 | 313.5 | 75.8 | 5.0 | 1113.6 |
| Rule | 34.5 | 356.3 | 33.7 | 22.6 | 346.2 | 76.2 | 0.0 | - |
| RL | 45.8 | 241.8 | 35.3 | 33.1 | 256.6 | 77.0 | 8.5 | 1044.9 |
| MCTS | 59.2 | **147.3** | **12.3** | 30.6 | **145.1** | 63.6 | 18.0 | 939.1 |
| LLM (Llama-13b) | 56.2 | 192.6 | 21.4 | 34.1 | 193.1 | 69.9 | 16.2 | 1090.1 |
| LLM (GPT-3.5) | 63.5 | 166.6 | 13.5 | **38.5** | 160.0 | 56.5 | 16.2 | 804.9 |
| LLM (GPT-4) | **67.7** | 158.5 | 16.1 | 38.2 | 153.8 | **51.3** | **33.9** | 555.8 |

**Quantitative results** According to the quantitative results in Table 1, the proposed HAZARD benchmark presents a significant challenge, as all the *Random*, *Rule*, and *Greedy* methods exhibit poor performance across all three scenarios. The *MCTS* method inherently reasons the environment changes through simulation and therefore performs the best among baseline methods. Surprisingly, although not finetuned on the training data, the LLM pipeline demonstrates superior performance compared to most baseline methods on three scenarios, showing its strong zero-shot decision-making capabilities. Furtherly, the results indicate a clear difference in decision making ability among different LLMs, as the GPT-4 model outperforms both GPT-3.5-turbo model and LLaMa-13b chat model by a large margin. Notably, in the wind scenario, all methods struggle with a remarkably low *Value* score, highlighting the difficulty of reasoning the movements of objects.

**Perceptional results** According to Table 1, all methods show reduced performance in the *with perception* scenario, highlighting the challenges of perception in dynamic environments. The perception model struggles to detect objects submerged in water or obscured by flames. Interestingly, in wind scenarios, this perceptual difficulty can be beneficial, as agents stop pursuing objects blown far away. Notably, despite these challenges, LLM-based agents still demonstrate competitive and robust decision-making ability.

**Qualitative results** As illustrated in Figure 5, the LLM pipeline shows the ability to take into account basic attributes during decision-making processes, enabling it to make rational choices in some cases. For instance, in the fire scenario, the LLM pipeline demonstrates comprehensive consideration of object temperature, distance, and value in the reasoning path. Accordingly, the LLM finally selects a valuable target with a low temperature, which is an optimal choice in this situation.

**Prompt:**
......
Objects states history:
Object id 75, object location: x 246.5, y 256.5, object distance from me is 16.57 m, object temperature is 56.66 Celsius at step 503, 71.29 Celsius at step 609.
Object id 84, object location: x 254.1, y 279.0, object distance from me is 25.6 m, object temperature is 41.78 Celsius at step 331, 51.66 Celsius at step 437, 54.41 Celsius at step 470.
Object id 87, object location: x 261.8, y 288.6, object distance from me is 33.62 m, object temperature is 38.77 Celsius at step 331, 45.24 Celsius at step 437, 49.97 Celsius at step 470.
......

**Reasoning Path:**
1. Evaluating the object "apple" (75): Although it is closest to you, its value is low (1) and its temperature is quite high (71.29 Celsius), indicating it might start burning soon.
2. Evaluating the object "bag" (84): Its value is higher (5), but it's farther away and its temperature is also high (54.41 Celsius), which might also start burning before you reach it.
3. Evaluating the object "bag" (87): Although it's the farthest, its temperature is the lowest (49.97 Celsius) and its value is high (5), so it might not start burning before you reach it.

**Egocentric View:**

**Final Outputs:**
go pick up object <bag> (87)

Figure 5: **A qualitative result of the LLM pipeline**. The GPT-4 model takes simple attributes, such as distance, temperature, and object value into consideration to enhance its decision-making abilities.

**Failure cases** In Figure 6, we provide two failure cases of the LLM pipeline. In the first case on the left, the LLM struggles with effectively considering dynamics during decision-making. The LLM pipeline chooses to walk towards the closest target, the backpack object. However, the object is swiftly carried away by the wind, leading to the failure of the LLM in catching up with the backpack object. In the second case on the right, the LLM pipeline suffers from inconsistency between its reasoning and prediction. Despite the thought process indicating that the optimal choice is to search for other target objects, which leads to hairbrush with id 66, it ultimately fails to select the desired target as its final decision.

**Prompt:**
I'm in an outdoor scene where an intense and turbulent wind is going on. I want to save as many valuable target objects as possible by picking them up and putting into some fixed shopping carts.
......
Target objects currently seen:
name: basket, id: 9, value: 1, distance: 24.83
name: backpack, id: 5, value: 5, distance: 14.21

**Reasoning Path:**
the best available action would be B. go pick up object <backpack> (5) and then try to put it into a shopping cart. If unsuccessful, we should try to look around for another shopping cart or hold onto the backpack until we find one.
**Action result:**
**Fail**

**Prompt:**
Suppose I am wearing fireproof suits and my house is on fire. I want to save as many valuable target objects as possible by picking them up and putting them into my bag.
...... Previous actions: go pick up object <hairbrush> (64) (paused after taking 100 steps)
Available actions:
A. go pick up object <hairbrush> (64)
B. go pick up object <hairbrush> (66)

**Reasoning Path:**
...... Therefore, it is not recommended to choose this action. Given the available actions, it seems that none of the hairbrushes are good options to save. **It may be more beneficial to search for other target objects that have not been previously attempted to be saved.**
**Final Outputs:**
**A. go pick up object <hairbrush> (64)**

Figure 6: **Two failure cases of the LLM pipeline**. The LLM pipeline will collapse when it fails to reason the environmental change that the backpack object is being blown away (on the left), or is inconsistent between reasoning and predicting (on the right).

# 6 CONCLUSION

We introduce HAZARD, a novel challenge with dynamically changing environments. To support environment changes, we develop a simulation system on top of the ThreeDWorld platform. This system includes a physical simulator and a visual effect generator, enabling simulations of fire, flood, and wind scenarios. Leveraging this framework, we design an object rescue task for embodied agents and generate a dataset for this task. Subsequently, we evaluate and analyze the performance of large language model (LLM) agents and existing baseline methods using the generated dataset. However, the HAZARD challenge focuses only on object rescue. In the future, we will introduce more actions to the simulator to allow agents to mitigate environmental changes (e.g., using an extinguisher to put out fires).

REPRODUCIBILITY STATEMENT

For readers interested in reproducing the experimental results presented in this paper, we have made our experiments accessible via a Github repository, available at `https://github.com/UMass-Foundation-Model/HAZARD`. For implementation details, please refer to the documentation within the repository.

ACKNOWLEDGMENT

We thank Tianmin Shu and Chuangchuang Sun for their insightful discussions, Dongyu Ji for helping us design the pipeline figure, Jeremy Schwartz and Esther Alter for setting up the ThreeDWorld environments. We thank the anonymous reviewers for their helpful suggestions. This work is funded in part by grants from ONR Science of AI Program, Google, Amazon, Cisco, and Toyota Motor North America.

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

## A EXPERIMENT DETAILS

**Navigation** An integrated navigation module based on perception and $A^*$ planning is provided to all agents. The module divides the world into grids of size $0.25$ and calculates the maximum height of objects inside each grid using RGB-D observation. It then assigns weights to each grid according to the exponential of height. Finally, it runs an $A^*$ path planning algorithm on the grids and controls agents to walk to the path midpoints sequentially. The module also provides a 'walk one step' option, in which the agent only walks to the first midpoint.

**Perception model details**  Since we used RCNN-based perception models, we map each detected instance to a ground truth object index during testing. For this purpose, we find the ground truth object in the same category as the detected instance and has the most overlapping bounding box with that instance, and then assign its index to the instance. Since we need to make sure the same instance is mapped to the same index across different frames, we have to use some of the ground truth information here. However, we keep GT information usage minimal.

**Random Agent**  We implemented an agent that does random actions chosen from:

- Walk one step closer to the nearest target object.
- Walk one step closer to the nearest container (available in 'wind' scene).
- Pick up the nearest target object.
- Drop the object in hand into the nearest container in 'wind' scene or into the grasped container on another hand in other scenes.
- Turn around and explore.
- Walk to a random object in sight.

**Reinforcement Learning**  Our RL agent uses the same set of actions as the random agent. We design the cumulative reward as the sum of the following terms

- For each object retrieved, add $20$ to the reward.
- If the agent is holding something, add $-10$ to the reward. This penalizes the agent for holding an object for too long.
- The distance from the agent to the nearest target or container, depending on whether it is grasping a target.
- For each action done, add $-0.1$ to the reward. For invalid actions, add $-5$ instead.

The agent is given a map of $4 \times W \times H$ where $W$ and $H$ are the sizes of the grid map set in advance. One channel contains binary information denoting if each grid is explored, while another contains height information as in navigation. We also provide the object id in each grid (if any) in the third channel and agent information in the last channel. For fire and flood scenes respectively, we additionally provide the temperature or water level in a separate channel.

We use the PPO algorithm with learning rate $2.5 \times 10^{-4}$ and train for $10^5$ steps. We did not run a typical $10^6$ steps for two reasons: low sample frequency due to physics simulation, and the model collapsing to doing only the explore action after sufficient steps. We use an early-stopped training result of our RL agent to do our evaluation.

**Rule-based Agent**  Another set of high-level motions is used in all agents except Random and RL, which is outlined as follows:

- **walk to nearest**: The agent moves towards the nearest object, with the heuristic cost equating to the distance required to reach the destination;
- **explore**: This action involves the agent exploring its surroundings to discover more objects, and the heuristic cost for this action is a constant;
- **pick up nearest**: When an agent is near an object, he can perform the action to pick the nearest object up;
- **drop**: The agent can drop the object in hand to a container or ground;

Our rule-based agent first decides all available actions from the objects visible from the observation or in memory. It then randomly selects a target object, walks to it, picks the target up, possibly walks to a random container, and drops it. This process is repeated until the step threshold is reached or no target object is visible. In the latter case, the agent does a 'look around' action and tries to select a target again.

My house is on fire now, I want to save as many valuable target objects as possible by picking them up and putting them into my bag. Given the current state and my previous actions, please help me choose the best available action to save as many valuable target objects as possible. All objects are denoted as <name> (id), such as <table> (712), and different objects have different values. Note objects lose their value once they start burning. The status of the objects previously seen may not be accurate any more. The final score is measured by the total value of the target objects I saved successfully.

Target objects:
name: bag, handbag, pocketbook, purse, value: 5, attribute: None
name: hairbrush, value: 1, attribute: None

Current State:

Target objects currently seen:
name: bag, handbag, pocketbook, purse, id: 56, value: 5, distance: 21.71, temperature: 41.81 Celsius
name: hairbrush, id: 57, value: 1, distance: 19.81, temperature: 42.43 Celsius

Target objects previously seen:
name: hairbrush, id: 64, value: 1, distance: 36.45, temperature: 56.49 Celsius
name: hairbrush, id: 66, value: 1, distance: 36.17, temperature: unknown

Previous actions:
go pick up object <bag, handbag, pocketbook, purse> (62) (success), look around (success), go pick up object <bag, handbag, pocketbook, purse> (56) (paused after taking 100 steps)

Objects states history:
Object id 56, object location: x 242.0, y 231.5, object temperature is 37.53 Celsius at step 277, 41.81 Celsius at step 383.
Object id 57, object location: x 249.33, y 233.67, object temperature is 37.51 Celsius at step 277, 42.43 Celsius at step 383.
Object id 64, object location: x 264.17, y 284.0, object temperature is 56.49 Celsius at step 43.

Available actions:
A. go pick up object <bag, handbag, pocketbook, purse> (56)
B. go pick up object <hairbrush> (57)
C. go pick up object <hairbrush> (64)
D. go pick up object <hairbrush> (66)
E. look around

Answer: Let's think step by step.

Figure 7: **An example of detailed prompt input for LLM**. The example is selected from a fire scene.

**MCTS Agent**  The MCTS agent applies the Monte Carlo Tree Search (MCTS) strategy for action planning and decision-making to rescue objects. It can memorize all the objects it has seen and simulate the object status transition in its mind before making a decision. When faced with a decision, the agent identifies a plan that results in the lowest heuristic cost to complete the task at hand. The first action of this plan is then selected as the current action. In detail, in each step, the agent will simulate 2000 times. To consider object properties and historical information, the MCTS agent retains past observations of objects, enabling linear predictions of their status and value in the next step. Subsequently, these object value predictions are incorporated into the heuristic cost calculation employed by MCTS.

For each decision in MCTS, the score $Q(s, a)$ and policy $\pi(s, a)$ for each action $a$ and current state $s$ is:

$$Q_{MCT}(s, a) = Q(s, a) + c(n(s)) \frac{\sqrt{n(s)}}{1 + n(s, a)}; \tag{1}$$

$$\pi_{MCT}(s) = \text{argmax}_a Q_{UCT}(s, a). \tag{2}$$

where $n(s), n(s, a)$ is the visiting times for state $s$ and action $a$ at state $s$ respectively, $Q(s, a)$ is the mean value of action $a$ at state $s$, $c(n(s))$ is a function related to $n(s)$:

$$c(x) = \log \left( \frac{1 + x + c_0}{c_0} \right) + c_1, \tag{3}$$

Here, $c_0 = 10^6$ and $c_1 = 0.1$.

**Greedy Agent**  We modify the selection of our rule-based agent to create the greedy agent. Instead of randomly choosing a target, it chooses the target or container that has the lowest heuristic cost calculated in the same way as the MCTS agent, to the best of the agent's knowledge (memory and observation). If no target object is available, we also let the agent look around.

**LLM-based Agent**  In Figure 7, we illustrate an examplar input prompt for the LLM-based agent. This prompt begins with an outline of the task description, prompt format, and agent's objective. The *Target objects* section enumerates the names, values, and attributes of the target objects. *Current state* depicts objects the agent holds, which is empty in this example as the agent currently holds nothing. The sections, *Target objects currently seen* and *Target objects previously seen*, represent target objects in the present and past semantic maps, respectively. *Previous actions* records prior agent actions, while *Objects states history* includes observed object states over time. The prompt concludes with a list of possible actions.

**Computational resources used**  We run most of our experiments on an Intel i9-9900k CPU and RTX2080-Super GPU Desktop. Each trial takes no more than 15 minutes, except for LLM-based agents where most of the time is spent on API calls. Our RL training took 20 GPU hours for each scene.

## B  EXPLORING THE IMPACT OF ENVIRONMENTAL EFFECTS ON AGENTS

Table 2: The performance of baseline methods when agents are affected by hazards on the 'without perception' version of HAZARD. The results in the fire scene is not included because agents can hardly complete the task when being affected by flames.

| Methods | Flood | | | Wind | |
|---|---|---|---|---|---|
| | Value↑ | Step↓ | Damage↓ | Value↑ | Step↓ |
| Greedy | 13.9 | 328.4 | 81.5 | 0.0 | - |
| Random | 28.9 | 293.4 | 74.3 | 4.6 | 1148.5 |
| Rule | 24.4 | 327.7 | 73.4 | 0.0 | - |
| RL | 33.2 | 258.9 | 64.5 | 13.1 | 890.6 |
| MCTS | 42.3 | 146.7 | 70.8 | 16.9 | 938.2 |

In the default setting of HAZARD, agents are not affected by environmental hazards, including fire, flood, and wind. To investigate how agents performs when they are impacted by environmental hazards, we provide an additional setting supported by HAZARD as follows:

- Fire: In the fire scene, the agent has its own temperature affected by fires. By limiting the agent's temperature upper bound, we keep the agent away from regions occupied by fire.

- Flood and Wind: In flood and wind scenes, agents are affected by flood or wind forces. These forces slow down the agents' actions when agents are moving against the direction of the flood or wind.

Table 2 shows the results in wind and flood scenes when baseline agents are influenced by environmental forces. Most of baseline methods have a minor decline in performance. From these results, we can infer that the impact of environmental factors marginally increases the difficulty level in both flood and wind scenes. The change in fire scenes requires all baseline methods to be modified, as fire blocking the way will greatly affect our path planning and decision making components. Therefore, we leave this part of experiments as future work.

## C  ADDITIONAL TEST SETS

Table 3: The performance of baseline methods on test set with new objects. We use fire and flood scenes with 'with perception' version of HAZARD for this evaluation.

| Methods | Fire | | | Flood | | |
|---------|------|------|--------|-------|------|--------|
| | Value↑ | Step↓ | Damage↓ | Value↑ | Step↓ | Damage↓ |
| Greedy | 31.3 | 326.5 | 32.2 | 23.6 | 318.8 | 80.0 |
| Random | 42.3 | 311.9 | 24.5 | 32.5 | 291.6 | 76.9 |
| Rule | 35.8 | 334.3 | 28.0 | 21.9 | 332.3 | 84.9 |
| RL | 45.7 | 291.9 | 25.5 | 36.0 | 233.0 | 77.2 |
| MCTS | 69.6 | 164.5 | 16.5 | 32.2 | 181.1 | 71.2 |

To assess the generalization capabilities of embodied agents more comprehensively, we have introduced the following two additional test sets:

1. Test Set with New Objects: This test set includes target objects that were not present in the training set.

2. Test Set with Larger Rooms: The rooms in this test set are larger than those in the training set.

For the test set with new objects, the 'without perception' version of HAZARD does not present additional challenges, as agents are provided with ground truth labels. In this section, we propose a method to tackle the test set with new objects in the 'with perception' version of HAZARD. We have replaced the R-CNN perception module with Grounded-SAM (Kirillov et al., 2023; Liu et al., 2023b) to generate segmentations for scenes containing new objects. As illustrated in Table 3, this powerful perception module helps all baseline models to retain most of their effectiveness when encountering new objects.

## D  TRAINING DEMONSTRATIONS GENERATION

To enable the training of imitation learning methods, we provide demonstrations on training sets using an oracle planner with access to complete ground truth information. After obtaining the ground truth information for all time steps, the oracle planner traverses all possible rescue plans and identifies the one with the highest value. If multiple plans have the same value, the plan with the fewest time steps is selected. However, this planner is imperfect because it assumes successful action execution (e.g., unobstructed navigation).

