# OpenReview forum: "HAZARD Challenge: Embodied Decision Making in Dynamically Changing Environments"
_ICLR.cc/2024/Conference — ICLR 2024 poster_

### Official Review · Reviewer_qCJU · 2023-10-29

**Soundness:** 3 good
**Presentation:** 3 good
**Contribution:** 3 good
**Rating:** 8
**Confidence:** 3

**Summary:**

This paper builds a novel embodied AI environment, addressing the dynamically changing environments. They also provide an API to employ large language models (LLMs) for action selection.

**Strengths:**

The ability to detect changes and adapt to changes in dynamically changing environments is key to intelligent agents. It is good to see people start to develop environments to address such challenges. The authors also provide support for LLMs to perform action selection.

**Weaknesses:**

see questions

**Questions:**

I am curious about how hard are the three environments. Also, which decision-making approaches do you think will dominate in the HAZARD challenge?

---

> ### Author Response · Authors · 2023-11-22
> **Reply to Reviewer qCJU**
>
> > How hard are the three environments
>
> The HAZARD challenge's difficulty primarily comes from its constantly changing environments. We believe it is challenging enough to effectively differentiate the performance of various methods. However, it is not excessively difficult, as all agents can complete the task to some degree. Specifically, the challenge involves planning in dynamic environments, comprehending the effects and dynamics of these changes, and accurately perceiving the evolving surroundings.
>
> > Which decision-making approaches do you think will dominate in the HAZARD challenge?
>
> Thank you for your question. Based on current results, LLM-based agents demonstrate strong decision-making capabilities, while MCTS-based agents excel among other baselines. The success of MCTS may be attributed to its natural ability to simulate environments, leading to a deeper understanding of potential future states. Therefore, we identify two key factors for superior performance in HAZARD:
> * Agents capable of simulating environmental changes and imagining future states accurately during decision-making, then basing their decisions on these internal simulations.
> * Agents powered by more robust foundational models. Our research reveals that LLM-based agents with a GPT-4 backbone outperform those using GPT-3.5, and GPT-3.5 surpasses Llama-13B. This trend indicates that LLM-based agents benefit from stronger foundational models.

---

### Official Review · Reviewer_BqkX · 2023-10-30

**Soundness:** 3 good
**Presentation:** 3 good
**Contribution:** 3 good
**Rating:** 8
**Confidence:** 4

**Summary:**

The paper proposes a new simulated benchmark for interactive agents that focuses on dynamic changes in the environment, caused by fire, flooding or strong winds, which the agent needs to react to dynamically. It provides a simulator, a set of 100 scenes for each of the three tasks (split between train and test) and a language interface that allows LLM-based agents to interact with the environment. Evaluations of LLM and non-LLM agents show that the common-sense reasoning in LLMs is beneficial, but they lack ability to react to dynamic environment changes.

**Strengths:**

The simulator is an interesting step towards testing agents' abilities in scenarios where the environment is dynamically changing and these changes are uncontrollable by the agent. This seems like a scenario of practical relevance for search-and-rescue applications.

The physics of the fire / flood / wind, albeit simplistic, seem sufficient for meaningful yet systematic changes to occur in the environments. The visual rendering of the effects is sufficiently realistic judging from the photos provided.

It is nice that the benchmark supports language-based descriptions and interaction out of the box to facilitate research on higher level reasoning systems while removing the burden of low-level control and perception if not desired.

Further, the procedural environment generation allows for the creation and evaluation of diverse scenes, so that overfitting to a few environments is prevented.

**Weaknesses:**

While I like the text-based interface, it also seems like a weakness of the benchmark that it seems primarily designed to evaluate the high-level reasoning capabilities of agents, rather than the potentially more challenging low-level manipulation aspects of these tasks. To put differently, would there be any difference if the benchmark wasn't implemented with a nice graphics pipeline but instead as a fully text-based game a la nethack? My understanding is that at least the main LLM planner method would work without change.

I do acknowledge that the authors try to provide versions of the benchmark that require visual perception of the environment. It would be nice though if a similar array of different options was provided on the action representation side. The current benchmark only supports very high-level actions. It is further unclear whether the RL policy that the authors compare to has all the same privileged observation and action primitive access that the LLM planner has. A fair comparison would be good here.

There is some information that is missing from the paper (see questions below). The paper does provide some videos on their website, but the videos "with agent" did not load for me, which makes it hard to judge how fast the environment changes with respect to the speed of the agent, and thus how challenging the tasks are.

While the current split of train:test is reasonable, it would be nice to investigate certain axis of generalization for the different agents more systematically. E.g. one could specifically test generalization to larger rooms or to new objects etc.

Finally, as acknowledged by the authors, the agent currently has no way of *influencing* the dynamics of the environment, e.g. by putting out the fires instead of working around them. It would be good to add such capabilities to more holistically evaluate disaster responses, but I acknowledge that this is challenging and the lack thereof does not invalidate the contributions of this submission.

**Questions:**

- is the robot agent itself incurring damages as it moves eg through fire / water?

- is there a way to generate training demonstrations, e.g. for an imitation learning pipeline?

- what simulation speed does the current simulator support? can it be run on headless servers & with multiple workers in parallel? this is important for understanding whether it can support RL workflows.


# Post-Rebuttal Comments

Thank you for answering my review. I appreciate the adaptations to the benchmark, in particular taking environment effects on the agent into account and supporting generalization evaluations.
Overall, I support acceptance of the paper. I also skimmed through the other reviews and the rebuttal seems to at least in part address the concerns of the reviewer that voted "marginally below acceptance", so I will increase my score to accept.

**Details Of Ethics Concerns:**

--

---

> ### Author Response · Authors · 2023-11-22
> **Reply to Reviewer BqkX**
>
> > Would there be any difference if the benchmark wasn't implemented with a nice graphics pipeline but instead as a fully text-based game a la nethack? My understanding is that at least the main LLM planner method would work without change…It would be nice though if a similar array of different options was provided on the action representation side. The current benchmark only supports very high-level actions.
>
> Thank you for your comment. Unlike text-based games, our challenge supports the simulation of continuous observations and low-level actions, such as moving a specific distance or turning by a precise degree.
>
> Following your suggestion, we have provided interfaces for using low-level action APIs for agents, along with relevant documentation. However, when limited to low-level actions alone, current agents struggle with task completion. For example, when we replace the action set of wind scenes with low level actions {turn, move forward, pick up, drop} instead of {walk to nearest, pickup nearest, drop}, the RL agent almost fails to retrieve even one single object.
>
> We preliminarily test the RL agent equipped with low-level actions in wind scenes. However, as the following results demonstrate, the agent almost fails in this setting, indicating that this new setting poses significant challenges to the agents.
>
> | Without Perception     | Value | Step   |
> |------------------------|-------|--------|
> | RL                     | 8.5   | 1044.9 |
> | RL (low-level actions) | 3.7   | 1144.0 |
>
> > It is further unclear whether the RL policy that the authors compare to has all the same privileged observation and action primitive access that the LLM planner has. A fair comparison would be good here.
>
> Yes, the observations are the same. We modify the observation to fit a 2D-array for RL and text description for other methods, but they all have access to full observation. We modified the actions of RL a bit as described in the paper, as it’s infeasible for RL to output exact object indices.
>
> > It would be nice to investigate certain axis of generalization for the different agents more systematically. E.g. one could specifically test generalization to larger rooms or to new objects etc.
>
> In response to your valuable suggestion, we have created two new test sets: one with new objects and another with larger rooms. We've also provided documentation to facilitate the addition of new objects.
>
> In addition, we propose a method to tackle the test set with new objects in the ‘with perception' version of HAZARD. We have replaced the R-CNN perception module with Grounded-SAM to generate segmentations for scenes containing new objects. We examine this new perception module on the test set with new objects, and results are summarized as follows:
>
> | With Perception | Fire  |       |        | Flood |       |        |
> |-----------------|-------|-------|--------|-------|-------|--------|
> |                 | Value | Step  | Damage | Value | Step  | Damage |
> | Rule            | 35.8  | 334.3 | 28.0   | 21.9  | 332.3 | 84.9   |
> | Greedy          | 31.3  | 326.5 | 32.3   | 23.6  | 318.8 | 80.0   |
> | Random          | 42.3  | 311.9 | 24.5   | 32.5  | 291.6 | 76.9   |
> | RL              | 45.7  | 291.9 | 25.5   | 36.0  | 233.0 | 77.2   |
> | MCTS            | 69.6  | 164.5 | 16.5   | 32.2  | 181.1 | 71.2   |
>
> According to the results, with the help of the strong segmentation model, all baseline models are able to retain most of their effectiveness when encountering new objects. The results are included in Appendix C.
>
> > The agent currently has no way of influencing the dynamics of the environment
>
> Thank you for your insightful comments. We are developing more diverse actions to enhance agents' interaction with the environment. For instance, we've introduced a 'put-out-fire' action, enabling an agent to extinguish fire at a specific location (please refer to the ‘put out fire’ gif on our website). We are also developing actions for water pumping and wind blocking. Such additions will not only increase mission complexity but also make the challenge more engaging.

---

> ### Author Response · Authors · 2023-11-22
> **Reply to Reviewer BqkX - part 2**
>
> > Is the robot agent itself incurring damages as it moves eg through fire / water?
>
> Currently not. We value your suggestion and have accordingly integrated the impact of hazards on agents into the HAZARD challenge. To achieve this, we've developed a new setting where agents are directly affected by environmental effects:
> * In fire scenes, the environment now causes damage to agents from fire by influencing the agent's temperature. This adaptation requires modifications to all baseline methods, particularly for fire scenes, as fire often blocks paths, demanding a different approach of planning algorithms.
> * For flood and wind scenes, we've introduced a setting where agents are influenced by drag forces. Detailed descriptions and results of baseline methods in this new setting can be found in Appendix B. Given that most baseline methods show only a slight performance decrease, it suggests that environmental factors slightly increase the challenge in both flood and wind scenes.
>
> > Is there a way to generate training demonstrations, e.g. for an imitation learning pipeline?
>
> We have produced demonstrations on a training set using an oracle planner with access to complete ground truth information. After obtaining the ground truth information for all time steps, the oracle planner traverses all possible rescue plans and identifies the one with the highest value. If multiple plans have the same value, the plan with the fewest time steps is selected. However, this planner assumes successful action execution (e.g., unobstructed navigation), so it may not be perfect. This part of discussion is included in Append D.
>
> > What simulation speed does the current simulator support? can it be run on headless servers & with multiple workers in parallel? This is important for understanding whether it can support RL workflows.
>
> The HAZARD benchmark, developed on the ThreeDWorld simulator, can accommodate multiple workers on a single machine and is compatible with headless servers equipped with an X11 server. For further details, please refer to the ThreeDWorld documentation (https://github.com/threedworld-mit/tdw/tree/master/Documentation). Slow simulation is a common challenge with many simulators, but it can be mitigated through a large pool of parallel processes or by setting a lower resolution.

---

### Official Review · Reviewer_oxZh · 2023-10-30

**Soundness:** 4 excellent
**Presentation:** 4 excellent
**Contribution:** 4 excellent
**Rating:** 6
**Confidence:** 4

**Summary:**

This paper presents a new virtual environment for embodied agents from the perspective of dynamic environmental change in the real world, focusing on unexpected disaster scenarios, including fires, floods, and winds. A benchmark HAZARD is developed to evaluate embodied agents making decisions in dynamically changing environments. Further qualitative results on HAZARD the challenges of dynamic environments to existing baseline agents. Notably, the limitations of reasoning and responding to dynamically changing environments for LLM-based agents are analyzed.

**Strengths:**

- This work simulates new scenarios of dynamic unexpected disasters for embodied agents. Based on the simplified physical model, the scenes achieve a good balance between realistic physical properties and real-time simulation
- This work provides a new simulated embodied benchmark HAZARD to save valuable items in unexpected disaster scenarios to assess the performance of agents in dynamic simulations. The new benchmark will promote the study and enhancement of the performance of embodied agents in dynamic scenarios
- Under the HAZARD benchmark, the quantitative results on a range of baselines, including LLM-based agents, provide a base performance of all baseline methods in dynamic environments.
- The paper proposes to focus on the ability of embodied agents to respond quickly to environmentally driven changes, and reveal the strengths and limitations of LLMs-based agents based on the results, where the performance of the LLMs-based approach is limited by the dynamic environment.
- The paper is well written and easy to follow with its detailed documentation and open source code.

**Weaknesses:**

- The rendering quality and motion effects of the simulated scene are still somewhat different from reality, which may result in some perceptually based agents using camera images that will lead to domain gaps.
- The current scenario exhibits a limited scale. Is there a method available to efficiently expand the current scenario, such as loading assets and assigning attributes to different disaster scenes?
- The proposed LLM pipeline has to input the same task description at each step to perform sequential decisions, which is not efficient.
- The mission objectives of the benchmarks are relatively simple, focusing on the task of object rescue, and the benchmarks can be expanded further in terms of mission complexity.

**Questions:**

See Weaknesses

---

> ### Author Response · Authors · 2023-11-22
> **Reply to Reviewer oxZh**
>
> > The rendering quality and motion effects of the simulated scene are still somewhat different from reality, which may result in some perceptually based agents using camera images that will lead to domain gaps.
>
> Thank you for your comments.
>
> We view rendering quality as a balance with simulation speed. Given the large scale of fire and flood in the HAZARD challenge, we abandon pursuit for fine-grained simulations, such as detailed fire particle effects and fluid simulation, to evaluate embodied agents more efficiently. Additionally, our work utilizes the ThreeDWorld simulator, which comparatively offers realistic visual effects among current simulators.
>
> > The current scenario exhibits a limited scale. Is there a method available to efficiently expand the current scenario, such as loading assets and assigning attributes to different disaster scenes?
>
> Yes. Our codebase includes documentation on efficiently adding new assets and assigning attributes within HAZARD. We have updated the documentation with more detailed instructions for loading assets and assigning attributes.
>
> > The proposed LLM pipeline has to input the same task description at each step to perform sequential decisions, which is not efficient.
>
> We apologize for any confusion. In the chat model, the chain-of-thought reasoning's second step only requires previous conversations and an added prompt stating, “Answer with only one best next action. So the answer is option". For decisions at different time steps, it is necessary to input the task description each time, as LLM-based agents struggle to make decisions without the description in their context.
>
> > The mission objectives of the benchmarks are relatively simple, focusing on the task of object rescue, and the benchmarks can be expanded further in terms of mission complexity.
>
> Thank you for your insightful comments. We are developing more diverse actions to enhance agents' interaction with the environment. For instance, we've introduced a 'put-out-fire' action, enabling an agent to extinguish fire at a specific location (please refer to the current ‘put out fire’ gif on our website). We are also developing actions for water pumping and wind blocking. Such additions will not only increase mission complexity but also make the challenge more engaging.
>
> Moreover, we wish to highlight that HAZARD's main difficulty lies in its dynamic environmental changes, which is our primary focus. Considering the baseline methods' performance on HAZARD, we believe the challenge already presents sufficient difficulty.

---

### Official Review · Reviewer_p2Ka · 2023-11-06

**Soundness:** 2 fair
**Presentation:** 3 good
**Contribution:** 2 fair
**Rating:** 5
**Confidence:** 3

**Summary:**

This paper proposed a new benchmark, ‘HAZARD’, to evaluate an agent's ability to complete a task in environments with changing dynamics. Specifically, the proposed environments simulate three different unexpected disaster scenarios:  fire, flood, and wind. The three disaster scenarios are simulated based on the ThreeDWorld platform. In addition, the authors evaluate several baselines, including an LLM-based approach, on the newly proposed benchmark. The experimental results show that the LLM-based approach outperforms other methods in most of the tasks.

**Strengths:**

Originality and Significance:
The reviewer found the proposed environments interesting. The environment could be valuable in two folds: (1) It evaluates an agent’s capability to adapt to changing dynamics, which is an important capability of an embodied agent. (2) The simulated three hazard scenarios could foster future research on rescuing embodied agents.


Quality:
The paper is technically sound. The proposed environments and LLM-based policy are thoroughly evaluated.


Clarity:
The paper is generally well-organized and easy to follow.

**Weaknesses:**

1. It seems that the hazard’s effect on the functionality of the embodied agents is not simulated. For instance, how does the fire and high temperature affect the functionality of the agent? Similarly, in the flood scenario, is the agent affected by the buoyancy force and drag force? The reviewer found the simulation of damage to the agents is an important piece of a realistic simulator.

2. How would the proposed approach address a partially broken embodied agent? Would approaches similar to Zeng [1] work?

3. The reviewer found the baselines in the experimental section somewhat weak. Comparing existing works, such as Landi [2] for embodied AI on changing environments could make the experimental section more convincing.

4. Could you elaborate why the MCTS-based method outperforms other baselines in terms of fire step and flood step?


[1] Moving Forward by Moving Backward: Embedding Action Impact over Action Semantics, Zeng et al., ICLR 23.

[2] Spot the Difference: A Novel Task for Embodied Agents in Changing Environments, Landi et al., 2022

**Questions:**

Please see the above section.

---

> ### Author Response · Authors · 2023-11-22
> **Reply to Reviewer p2Ka**
>
> > It seems that the hazard’s effect on the functionality of the embodied agents is not simulated.
>
> We value your suggestion and have accordingly integrated the impact of hazards on agents into the HAZARD challenge. To achieve this, we've developed a new setting where agents are directly affected by environmental effects:
>
> * In fire scenes, the environment now causes damage to agents from fire by influencing the agent's temperature. This adaptation requires modifications to all baseline methods, particularly for fire scenes, as fire often blocks paths, demanding a different approach of planning algorithms.
> * For flood and wind scenes, we've introduced a setting where agents are influenced by drag forces. Detailed descriptions and results of baseline methods in this new setting can be found in Appendix B. Given that most baseline methods show only a slight performance decrease, it suggests that environmental factors slightly increase the challenge in both flood and wind scenes.
>
> > How would the proposed approach address a partially broken embodied agent? Would approaches similar to Zeng [1] work?
>
> Currently, our approach to hazards can involve blocking paths or applying additional forces on agents, which do not result in partially broken embodied agents. Zeng [1]'s work examines different agent action impacts, distinct from our study of environmental change impacts on agent decisions. We have incorporated this discussion into our related work section.
>
> > The reviewer found the baselines in the experimental section somewhat weak. Comparing existing works, such as Landi [2] for embodied AI on changing environments could make the experimental section more convincing.
>
> Following your advice, we have introduced a new baseline, MCTS-diff, building upon the previously strongest baseline, MCTS. MCTS-diff adopts the concept from Landi's work, encouraging agents to move towards areas where environmental changes are more probable. This is implemented by assigning additional value to MCTS nodes with significant temperature or flood signal changes, guiding agents to areas with rapid environmental changes.
>
> However, Landi's work primarily focuses on identifying differences between two scenes, leading their method to motivate agents towards areas with probable environmental changes. In contrast, our environment has continuous changes. Our objective is to evaluate agent decisions under these ongoing environmental changes, rather than merely identifying the changes. As the following results indicate, MCTS-diff performs worse than MCTS. This suggests that exploring areas with environmental changes may not be an effective strategy in the HAZARD challenge.
>
> |           | Fire  |       |        | Flood |       |        |
> |-----------|-------|-------|--------|-------|-------|--------|
> |           | Value | Step  | Damage | Value | Step  | Damage |
> | MCTS      | 75.9  | 150.1 | 19.7   | 43.7  | 146.6 | 69.9   |
> | MCTS-diff | 72.2  | 156.6 | 20.9   | 40.6  | 150.0 | 73.2   |
>
> The summarization of this discussion has been included in our related work section.
>
> > Could you elaborate why the MCTS-based method outperforms other baselines in terms of fire step and flood step?
>
> The MCTS method outperforms other baselines in both Step and Value metrics. Compared to LLM-based agents, MCTS-based methods consistently pursue the target with the lowest cost without altering their target during navigation or replan chances. On the other hand, LLM-based agents might choose a different target upon reaching the maximum navigation steps and having the opportunity to replan. Consequently, MCTS agents typically achieve a lower Step count than LLM-based agents.

---

### Author Response · Authors · 2023-11-22
**General Response**

We sincerely appreciate the time and effort by all the reviewers in reviewing our paper. We are delighted to find that the reviewers have generally acknowledged our contributions:

* Topic: HAZARD focuses on dynamically changing environments, which not only enhances important capabilities of embodied agents but is also closely aligned with practical applications. [p2Ka, oxZh, BqkX, qCJU]
* LLM Support: We provide convenient support for LLMs as agents and an in-depth analysis of current LLMs' performance. [p2Ka, oxZh]
* Diverse Scenes: HAZARD supports the creation and evaluation of a variety of scenes. [BqkX]
* Future Impact: HAZARD can advance research in rescue agents [p2Ka] and dynamic scenarios. [oxZh]

And we also thank all reviewers for their insightful and constructive suggestions, which help a lot in further improving our paper. In addition to the pointwise responses below, we summarize some of the supporting updates in our paper according to reviewers’ suggestions.

* We have introduced APIs for agents to perform low-level actions, such as moving or turning by any degree. [Reviewer BqkX Q1]
* Two new test sets have been added: one with new objects and another with larger rooms, to assess the generalization capabilities of agents (in Appendix C). We also update the document for adding objects and assigning attributes. [Reviewer BqkX Q3, Reviewer oxZh Q2]
* We provide a new setting in which agents are influenced by fire, flood, and wind effects. We have included the discussion and results of this new setting in Appendix B. [Reviewer p2Ka Q1, Reviewer BqkX Q4]

We have revised and uploaded the supplementary materials and main paper. We hope our pointwise response below could clarify reviewers' concerns. We thank all reviewers' time again.

---

### Author Response · Authors · 2023-11-22
**Thank you and we are looking forward to your post-rebuttal feedback!**

Dear AC and all reviewers:

Thanks again for all the insightful comments and advice, which helped us improve the paper's quality and clarity.

The discussion phase is about to end soon and we kindly remind the post-rebuttal responses.

We would love to convince you of the merits of the paper. Please do not hesitate to let us know if there are any additional experiments or clarification that we can offer to make the paper better. We appreciate your comments and advice.

Best,

Author

---

### Meta-Review · Area_Chair_E2SN · 2023-12-17

**Metareview:**

The authors introduce a new environment for agents that involves environmental effects (e.g. fire, water, and wind).  The importance of the work is the introduction of a dynamic environment in which to study agents.  This produces novel planning challenges with a bit of realism with relevant dimensions to explore.

The setting is very synthetic.  Arguably, the critical challenges for hazards will be related to low level control, effects of damage to the agent, and so forth.  These are acknowledged by the authors and they add a small amount of this functionality to their new results in the appendix.

Overall, reviewers felt that the contributions were sufficiently meritorious for acceptance

**Justification For Why Not Higher Score:**

The number of scenarios is restrictive and even with the new APIs, there are still several open questions about the impact and generalization from the provided benchmark.

**Justification For Why Not Lower Score:**

The novelty of the problem, the environment in which to explore it, the insights from baselines.

---

### Decision · Program_Chairs · 2024-01-16

Accept (poster)